# Thermogenic Activation Downregulates High Mitophagy Rate in Human Masked and Mature Beige Adipocytes

**DOI:** 10.3390/ijms21186640

**Published:** 2020-09-10

**Authors:** Mária Szatmári-Tóth, Abhirup Shaw, István Csomós, Gábor Mocsár, Pamela Fischer-Posovszky, Martin Wabitsch, Zoltán Balajthy, Cecília Lányi, Ferenc Győry, Endre Kristóf, László Fésüs

**Affiliations:** 1Laboratory of Cell Biochemistry, Department of Biochemistry and Molecular Biology, Faculty of Medicine, University of Debrecen, H-4032 Debrecen, Hungary; szatmari-toth.maria@med.unideb.hu (M.S.-T.); abhirup.shaw@med.unideb.hu (A.S.); balajthy@med.unideb.hu (Z.B.); kristof.endre@med.unideb.hu (E.K.); 2Doctoral School of Molecular Cell and Immune Biology, University of Debrecen, H-4032 Debrecen, Hungary; 3Department of Biophysics and Cell Biology, Faculty of Medicine, University of Debrecen, H-4032 Debrecen, Hungary; csomos.istvan@med.unideb.hu (I.C.); mocsgab@med.unideb.hu (G.M.); 4Division of Pediatric Endocrinology and Diabetes, University Medical Center Ulm, D-89075 Ulm, Germany; pamela.fischer-posovszky@uniklinik-ulm.de (P.F.-P.); martin.wabitsch@uniklinik-ulm.de (M.W.); 5Laser Clinic, H-1012 Budapest, Hungary; lezerklinika@gmail.com; 6Department of Surgery, Faculty of Medicine, University of Debrecen, H-4032 Debrecen, Hungary; fgyory@freemail.hu

**Keywords:** beige adipocytes, UCP1, mitophagy, LC3, cAMP, Parkin

## Abstract

Thermogenic brown and beige adipocytes oxidize metabolic substrates producing heat, mainly by the mitochondrial uncoupling protein UCP1, and can thus counteract obesity. Masked beige adipocytes possess white adipocyte-like morphology, but can be made thermogenic by adrenergic stimuli. We investigated the regulation of mitophagy upon thermogenic activation of human masked and mature beige adipocytes. Human primary abdominal subcutaneous adipose-derived stromal cells (hASCs) and Simpson–Golabi–Behmel syndrome (SGBS) preadipocytes were differentiated to white and beige adipocytes, then their cAMP-induced thermogenic potential was assessed by detecting increased expressions of UCP1, mitochondrial DNA content and respiratory chain complex subunits. cAMP increased the thermogenic potential of white adipocytes similarly to beige ones, indicating the presence of a masked beige population. In unstimulated conditions, a high autophagic flux and mitophagy rates (demonstrated by LC3 punctae and TOM20 co-immunostaining) were observed in white adipocytes, while these were lower in beige adipocytes. Silencing and gene expression experiments showed that the ongoing mitophagy was Parkin-independent. cAMP treatment led to the downregulation of mitophagy through PKA in both types of adipocytes, resulting in more fragmented mitochondria and increased UCP1 levels. Our data indicates that mitophagy is repressed upon encountering a short-term adrenergic stimulus, as a fast regulatory mechanism to provide high mitochondrial content for thermogenesis.

## 1. Introduction

Emerging evidence suggests the therapeutic potential of thermogenic classical brown and beige adipocytes against obesity and obesity-related diseases, because they can dissipate energy to heat, thus providing effort-free weight loss [1,2]. Heat production is mainly mediated by the function of uncoupling protein-1 (UCP1), an inner mitochondrial membrane protein that facilitates proton leak and the uncoupling of the mitochondrial respiratory chain from ATP synthesis [3].

White adipocytes are specialized to the long-term storage of energy, contain a single large lipid droplet and few mitochondria, and are characterized by low UCP1 expression [4]. Classical brown and beige adipocytes, which were first characterized in rodents, have similar morphological and biochemical characteristics, including high mitochondrial content, multilocular small lipid droplets and high levels of UCP1 expression [5,6]. In adult humans, brown adipose tissue (BAT) is localized in six anatomical regions and constitutes up to 1.5% of the total body mass [7]. The thermogenic activity of BAT can amount to 5% of the basal metabolic rate, and can oxidize around 4 kg fat per year in adults [8]. The inducible beige adipocytes, which have the same origin as the white adipocytes, can be found interspersed in subcutaneous white adipose tissue (WAT) deposits [6]. The ratio of beige and white adipocyte precursors is determined during the early differentiation of mesenchymal progenitors, and is strongly influenced by genetic predisposition [9]. The biogenesis of beige adipocytes is induced in response to various stimuli, such as chronic cold exposure, physical exercise, PPARγ stimulation or β-adrenergic cue, which results in norepinephrine (NE) release from the sympathetic nervous system. This process is often called “browning” [10]. BAT constitutively expresses UCP1; however, for beige adipocytes, a continuous adrenergic stimulus is required to maintain high UCP1 expression and large amounts of mitochondria. Increasing evidence suggests that a large proportion of beige cells become “masked” with white adipocyte-like morphology when the thermogenic stimulus subsides. Then, masked beige adipocytes are able to quickly re-induce or “activate” their thermogenic program upon cold exposure or β3-adrenergic receptor (β3-AR) agonist treatment [11]. However, the regulatory mechanism of beige adipocyte inducibility and maintenance needs to be elucidated, especially in humans.

Mitochondria play a key role in the thermogenic and metabolic functions of brown and beige adipocytes. These dynamic organelles change their morphology during fusion and fission events; their shape can be shifted between larger networks of elongated mitochondria and small, fragmented units, respectively [12]. A high rate of fragmentation is a physiological response to adrenergic stimulation in rodent brown adipocytes, which enhances mitochondrial uncoupling and energy expenditure [12,13]. In human adipocytes, UCP1-positive mitochondria are mostly fragmented [14]. The mitochondrial turnover is essential in maintaining cellular homeostasis, and is regulated by a balance between biogenesis and degradation [15]. Several nuclear-coded transcriptional regulators, such as peroxisome proliferator–activated receptor γ (PPARγ) coactivator 1α (PGC-1α), nuclear respiratory factor (Nrf) 1 and Nrf2, and mitochondrial transcription factor A (Tfam), are responsible for the precise control of mitochondrial biogenesis [16]. The elimination of dysfunctional mitochondria is mediated through selective autophagy, termed mitophagy [17].

Autophagy is an evolutionarily conserved intracellular digestion and recycling process for unwanted or damaged cellular components in eukaryotic cells. During autophagy, a portion of the cytoplasm and the organelles are enclosed in autophagosome, a double-membrane vesicle, for delivery to the lysosome to be degraded [18]. Autophagosome biogenesis is orchestrated by many highly conserved proteins, encoded by autophagy-related genes (ATG). The main step of autophagosome formation is the conjugation of microtubule-associated protein 1 light chain 3 (LC3-I) to the phosphatidylethanolamine (PE), to generate the lipidated LC3-II, which is then inserted into the autophagosomes. The most commonly used autophagosome marker is the LC3, because the amount of LC3-II correlates with the number of autophagosomes [19].

Mitophagy is a quality control mechanism for maintaining a healthy and functional mitochondrial network [20]. It can be triggered by the reduction of mitochondrial membrane potential, resulting in the stabilization of PINK1 (PTEN (phosphatase and tensin homolog)-induced putative kinase 1) on the outer mitochondrial membrane (OMM), followed by the recruitment of Parkin, which ubiquitinates OMM proteins [21,22]. Selective autophagy adaptor proteins, such as NDP52 (*nuclear dot protein 52 kDa*), OPTN (*optineurin*), NBR1 (*neighbour of Brca1 gene 1*) and p62 (*SQSTM1*), act as bridges between the ubiquitinated mitochondrial proteins and LC3 to mediate the engulfment of mitochondria into autophagosomes [23]. In addition, mitophagy can also take place independently of ubiquitination and adapter proteins through the direct interaction of LC3 with mitochondria-localized proteins, such as BNIP3 (*BCL2/E1B 19 kDa-interacting protein 3*), NIX (*BNIP3 homology NIP3-like protein X*) or FUNDC1 (*FUN14 domain-containing protein1*) [24]. The involvement of mitophagy in differentiation and developmental processes has been demonstrated, for example in the elimination of mitochondria during reticulocyte maturation [25,26].

Recent intensive research on the molecular regulation of autophagy has revealed its importance in adipocyte biology. In rodents, Parkin-mediated mitophagy contributes to the conversion of beige adipocytes to white after the withdrawal of the β-adrenergic stimulus, which results in the loss of UCP1 expression and thermogenic capacity of these adipocytes [27,28]. Studies in the field of browning are widespread in rodents but limited in humans. Mouse and human adipocytes differ in quantity and location in different periods of life; hence it is essential to investigate the mechanism of browning in human adipocytes. The regulation of autophagy/mitophagy upon thermogenic induction in human adipocytes is yet to be investigated. Here, we describe how in human masked and mature beige adipocytes, a portion of mitochondrial mass is continuously removed by mitophagy, which is promptly downregulated by a thermogenic cue, suggesting the existence of a fast adaptation mechanism responding to cold or other browning stimuli.

## 2. Results

### 2.1. cAMP-Mediated Thermogenic Stimulus Results in Increased UCP1 Expression and Mitochondrial Content in Differentiated White and Beige Adipocytes

Recently, our research group successfully optimized [29,30] the previously described white [31] and brown [32] adipogenic differentiation protocols for human primary abdominal subcutaneous adipose-derived stromal cells (hASCs), to gain thermogenic beige adipocytes from the stromal-vascular fraction (SVF) of abdominal subcutaneous fat. The functional capacity of differentiated primary, and also human Simpson–Golabi–Behmel syndrome (SGBS), beige cells was shown by the typically increasing oxygen consumption (OCR) rate of both white and beige adipocytes in response to cAMP stimulus, as demonstrated by our published results [29,30] (Appendix A). Since SVFs contain both white and beige precursors [9], the white protocol results in both white and masked beige adipocytes, and the latter respond to thermogenic stimuli [29]. As expected, in the present study we found that UCP1 gene expression was significantly increased in hASCs after both differentiation protocols, with higher levels of UCP1 protein in the beige cells (Figure 1A,B). In addition to primary adipocytes, we examined a non-immortalized human adipocyte cell line model, namely the SGBS cell strain, to exclude the potential effect of heterogeneity and donor dependency [33]. This in vitro cell line model is comparable to subcutaneous hASCs, and capable of differentiating into functional and sustainable white [34] and beige adipocytes [30]. It provides an appropriate tool to directly examine thermogenic competency and activation (Appendix A). In the unstimulated condition, UCP1 gene expression was significantly higher in beige as compared to white SGBS adipocytes (Figure 1A).

To mimic adrenergic stimulation, we treated the differentiated adipocytes with the cell-permeable cAMP analogue dibutyryl-cAMP, and observed that cAMP significantly induced UCP1 mRNA levels in white and beige adipocytes in a time-dependent manner (Figure 1A). Moreover, it also significantly increased the UCP1 protein level in both types of primary and SGBS adipocytes, up to 14 h. The white SGBS adipocytes responded to thermogenic induction to a lesser extent than the beige ones, while the response was similar in hASC-derived adipocytes. UCP1 protein expression remained unchanged upon blocking lysosomal degradation [35] by chloroquine (CQ) (Figure 1B). In general, CQ treatment did not affect the expression of any of the investigated genes in this study (data not shown).

Next, we intended to examine changes in mitochondrial content as a result of thermogenic stimulus. The expression of PGC1α, a mitochondrial biogenesis regulator [16], was significantly elevated in response to cAMP in white and beige primary as well as SGBS adipocytes (Figure 1C). We found a significantly higher mitochondrial DNA content in beige as compared to white adipocytes, both in primary and SGBS cells, which was further enhanced by 6 hr cAMP treatment (Figure 1D). The protein levels of the mitochondrial respiratory chain complexes II and V were elevated in beige hASC-derived adipocytes. The thermogenic stimuli led to an increasing trend in the protein expressions of complexes I, II, IV and V in primary white adipocytes, and the complex I and III levels were moderately higher in primary beige adipocytes. In SGBS cells, the level of complex V was significantly higher in beige adipocytes under unstimulated conditions, and cAMP treatment resulted in elevated protein contents of the complexes I, II, III and V in white adipocytes, and complex IV in beige SGBS cells. CQ treatment did not influence the expression of oxidative phosphorylation complex proteins (Appendix A). Collectively, these data confirmed that beige differentiation led to elevated expressions of UCP1, as well as higher contents of mtDNA and some of the mitochondrial complexes as compared to the white protocol, and these could be further enhanced by cAMP stimulation in both primary and SGBS cells. However, higher mitochondrial content can also be induced by cAMP in both primary and SGBS-derived white adipocyte populations, presumably through targeting the masked beige cell population.

### 2.2. Thermogenic Activation Represses Autophagy

Recent studies in mice showed that the autophagy-dependent mitochondrial clearance was low in beige adipocytes, which then became upregulated during whitening upon removal of the β3-AR agonist [27]. We examined how cAMP-stimulus influenced the expression of selected genes that are essential for autophagosome formation. We found that the mRNA expression levels of ATG5, ATG7 and ATG12 were significantly decreased in a time-dependent manner upon thermogenic activation in white and beige primary and SGBS cells (Figure 2A,B).

Next, we performed immunoblotting to determine the expression of the specific autophagy marker proteins LC3 and p62. The detection of LC3-I’s conversion to LC3-II by western blot is a well-accepted method to monitor autophagic activity; an increased LC3-II/LC3-I ratio indicates a high level of ongoing autophagy [36]. p62 (*SQSTM1*) is a well-described autophagy adapter protein, which functions as a link between LC3 and the ubiquitinated targets. It is continuously degraded during autophagy, meaning that a decreased level of p62 can be observed at high autophagic activity. In contrast, the accumulation of p62 indicates the inhibition of autophagy [37]. To measure autophagic flux, we applied CQ, which by neutralizing intralysosomal pH, inhibits the last stage of autophagy, and the degradation of autophagic cargo as well as LC3-II [35]. Under basal conditions, a high LC3-II/LC3-I ratio and low p62 levels were detected in white (masked beige) hASC-derived adipocytes by immunoblotting. CQ treatment enhanced the level of LC3-II, which indicates a high rate of autophagic flux. Unstimulated primary beige adipocytes had lower autophagic activity as compared to masked beige (white) cells. As a result of cAMP-mediated thermogenic stimulus, the LC3-II/LC3-I ratio was decreased and the p62 level was increased in both types of adipocytes, indicating the repression of autophagy (Figure 2C). The effect of thermogenic stimulus on autophagy was similar in white and beige SGBS cells (Figure 2D). Altogether, these data provide evidence that human white (masked beige) adipocytes show high autophagic activity and flux, which is rapidly downregulated as a result of cAMP-mediated thermogenic stimulus.

### 2.3. cAMP Stimulation Results in Reduced Mitophagy and Increased Mitochondrial Fragmentation in Masked and Mature Beige Adipocytes

In the next step, we performed co-immunostaining with antibodies against LC3 to visualize autophagosomes and TOM20 (translocase of outer mitochondrial membrane 20), in order to follow the mitochondrial morphology and the co-localization of the mitochondrial and autophagy markers, which is a method commonly used to evaluate mitophagy [38].

Under basal conditions, a high number of scattered LC3 punctae was observed in the cytosol of white and beige primary and SGBS cells. CQ treatment caused a significant increase in the number of LC3 punctae per cell in hASC-derived adipocytes, indicating a high rate of autophagic flux. In response to a cAMP-stimulus, the number of LC3 punctae was reduced, suggesting the repression of autophagy/mitophagy in both types of differentiated hASC-derived and SGBS cells (Figure 3A,B).

We used the confocal images to quantify the co-localization of LC3 and TOM20 by calculating correlation between the pixel intensities of the two detection channels [39,40]. We observed that mitochondria co-localized with autophagosomes in unstimulated white and beige primary adipocytes (Figure 3A,B); the Pearson correlation co-efficient (PCC) values were 0.27 ± 0.09 and 0.32 ± 0.07, respectively, indicating ongoing mitophagy. Blocking the late step of the autophagic pathway by CQ led to increased co-localization (PCC values 0.33 ± 0.06 and 0.38 ± 0.11) in white and beige hASC-derived adipocytes, respectively, indicating an enhanced mitophagic flux. cAMP treatment led to a significant reduction in co-localization in white (PCC value 0.17 ± 0.08) and beige primary adipocytes (PCC value 0.27 ± 0.11), showing the reduction of mitophagy in response to thermogenic stimuli (Figure 3A). SGBS cells had similar tendencies of decreasing LC3 and TOM20 co-localization as a result of cAMP treatment (Figure 3B).

The unstimulated white and beige primary and SGBS cells contained mixed mitochondrial populations. In confocal images, there were significantly higher proportions of elongated forms compared to forms with fragmented morphologies. We observed that the administration of cAMP resulted in an increased number of small, rounded, fragmented mitochondria in hASC-derived white and beige, as well as in white SGBS, cells. CQ had the same effect (Figure 3A,B, Appendix A). We applied technical controls for immunostaining (secondary antibody negative controls), and mitochondrial damage was induced by FCCP treatment for observing enhanced mitophagy as a positive control (Appendix A).

These data collectively suggest that in unstimulated adipocytes, most of the small, fragmented mitochondria are removed due to the high level of mitophagy/mitophagic flux, and cAMP-mediated thermogenic stimulation leads to the repression of mitophagy, and accordingly the retention of more fragmented mitochondria.

### 2.4. Downregulation of Mitophagy upon Thermogenic Stimulus Is Mediated through PKA Activity

To clarify whether PKA contributes to the control of mitophagy in the thermogenically active human white (masked beige) and mature beige adipocytes, we applied pre-treatment with the chemical PKA inhibitor H89 before cAMP-stimulation. The cAMP-mediated increase in UCP1 expression was significantly reduced in white (masked beige) primary and SGBS cells by the blocking of PKA signaling, as expected (Figure 4A). The repressive effect of cAMP on autophagy markers was greatly blunted, the LC3-II protein level was increased, and p62 decreased when white and beige hASC-derived as well as SGBS cells were pre-treated with H89 and then stimulated with cAMP, as compared to the thermogenic stimulus alone (Figure 4B,C). These findings demonstrate that PKA signaling controls the cAMP-mediated downregulation of autophagic degradative activity in human beige adipocytes.

### 2.5. Thermogenic Stimulus Downregulates the Expression of Parkin and Other Mitophagy Related Genes but Silencing of Parkin Does Not Result in Higher Thermogenic Potential

Parkin-dependent mitophagy is the most extensively studied mitophagy pathway. Parkin is an E3 ubiquitin ligase, encoded by the PARK2 gene in humans and responsible for initiating the removal of damaged, senescent or unwanted mitochondria. The cell- or tissue-specific functions of Parkin-mediated mitophagy were not fully characterized yet [41,42]. A robust decrease in the PARK2 gene expression was observed upon cAMP treatment in both white and beige adipocytes (Figure 5A). Normalized Parkin protein expression was decreased significantly in beige primary adipocytes and white SGBS cells upon cAMP treatment (Figure 5B). Significant reduction was observed in the expression of prominent mitophagy adapters involved in the Parkin-dependent pathway, namely OPTN (optineurin) and CALCOCO (NDP52), in both white and beige primary and SGBS adipocytes (Figure 5C,D). These results point towards Parkin as a major player in the ongoing mitophagy in adipocytes.

Therefore, we silenced Parkin in differentiated adipocytes and investigated whether its downregulation resulted in a thermogenic phenotype. A siRNA-based approach was utilized and more than 60% reduction in Parkin protein expression was achieved in both white and beige primary adipocytes, which remained unaffected by cAMP treatment (Figure 6A). LC3-II/LC3-I ratio and p62 remained unchanged after Parkin silencing as compared to control siRNA, which suggests ongoing mitophagy at the same level (Figure 6B,C). Parkin silencing resulted in increased levels of UCP1 and certain mitochondrial complex subunit proteins (complex I, III and V) only in beige primary adipocytes. cAMP could similarly upregulate UCP1 protein expression in cells incubated with Parkin and control siRNA, indicating that Parkin deficiency did not lead to the appearance of a more active beige phenotype (Figure 6D,E).

Our findings led us to examine the expression of Parkin-independent mitophagy genes, such as BCL2L13, FKBP8, FUNDC1, BNIP3 and BNIP3L [41,43]. cAMP promptly downregulated the gene expression of BCL2L13, FKBP8 and FUNDC1 in both types of primary and SGBS adipocytes, and BNIP3 only in primary adipocytes (Appendix A). The expression of BNIP3L (NIX) showed an increasing trend upon cAMP treatment in both primary and SGBS cells, but the changes were not statistically significant (Appendix A). These results indicate a complex response to adrenergic stimuli in white and beige adipocytes, involving both parkin-dependent and independent mitophagy pathways.

## 3. Discussion

Here we show that in human masked and mature beige adipocytes, a portion of mitochondria is actively eliminated through mitophagy, which is rapidly repressed as a result of cAMP-mediated thermogenic stimulus. These results raise the possibility that the inhibition of autophagy/mitophagy may open a potential way for activating thermogenesis without β-adrenergic stimuli.

Beige adipocytes are derived from the same mesenchymal precursors as the white ones, and exist in a masked condition in subcutaneous WAT depots in the absence of adrenergic stimulation [11]. Claussnitzer et al. showed that specific genetic factors influence the early differentiation of mesenchymal adipocyte progenitors into white or beige cells [9]. Consequently, we have expected that part of the adipocyte population differentiates into masked beige cells when the white differentiation protocol is applied. Recently, our research group examined the functional capacity of differentiated hASCs and SGBS cells by mitochondrial OCR measurements. We demonstrated that when a white differentiation protocol was used, the adipocytes were able to increase their OCR and to induce their respiratory chain activity in response to cAMP stimulation, indicating that the existing masked beige cells could be activated [29,30]. When we applied the white differentiation cocktail during the presented experiments, signs of active mitochondrial biogenesis and increased mitochondrial content were found as a result of thermogenic stimuli, which was characterized by higher PGC1α gene expression and mtDNA content (Figure 1C,D). In addition, an increased protein level of respiratory complex components was also observed (Appendix A). It has been reported that adrenergic stimulus leads to transition in mitochondrial morphology, from an elongated form to small and spherical fragmented mitochondria, in the brown adipocytes of mice; this process promotes uncoupled respiration and increases energy expenditure, suggesting that changes in the mitochondrial architecture are required for thermogenic activation [13]. Here, we could demonstrate the appearance of a high proportion of fragmented mitochondria upon cAMP treatment in human masked beige and mature beige adipocytes (Figure 3A,B).

Elevated mitochondrial content with fragmented morphology can result from a decreased rate of mitophagy. Altshuler-Keylin et al. identified autophagy-mediated mitochondrial removal as a previously uncovered mechanism for beige adipocyte maintenance and energy expenditure in mice; they demonstrated that the inhibition of autophagy by pharmacological or genetic approaches can prevent the loss of thermogenically active beige adipocytes [27]. Here, we investigated the opposite direction of signaling, that is, how autophagy/mitophagy is regulated during thermogenic activation of masked beige human adipocytes. We found high autophagy/mitophagy rates and flux in human white adipocytes. In addition, our data provide direct evidence of the fast downregulation of this activity upon short-term cAMP treatment in beige adipocytes, such as the decreased mRNA level of ATG genes, the low LC3 II/LC3 I ratio, the high protein level of p62 (Figure 2A–D), the reduced number of LC3 punctae, the higher number of fragmented mitochondria, and the decreased co-localization of LC3 and TOM20 (Figure 3A,B). In parallel with the repression of autophagy, an increased UCP1 level and mitochondrial content were observed (Figure 1A–D), suggesting that this fast regulatory mechanism contributes to preserving cellular components that are necessary for effective thermogenesis. Our results for human beige adipocytes are consistent with the recent study in mouse BAT, claiming that the regulation of autophagy/mitophagy and thermogenesis are inverse processes [44,45].

When we applied the brown differentiation protocol, which led to the appearance of beige adipocytes without the addition of cAMP, the autophagy/mitophagy activity was lower than in white adipocytes (Figure 2C,D), resulting in a higher mitochondrial content and increased expression of UCP1 (Figure 1A–D). We observed the inhibitory effect of cAMP stimulus on mitophagy even in these differentiated beige cells (Figure 2A–D and Figure 3A,B), which suggests that there is an ongoing, relatively fast mitochondrial turnover even in mature beige adipocytes, providing a reserve pool of mitochondria for the immediate need of induced thermogenesis.

The binding of NE to β3-ARs increases the intracellular levels of cAMP, leading to the activation of the cAMP-dependent PKA signaling pathway, which induces the transcription of mitochondrial biogenesis and thermogenesis-related genes [5,46]. Another effect of PKA has been described in response to β3-AR stimulation; that it can inhibit autophagy through various mechanisms in mammalian cells. PKA can phosphorylate LC3 directly [47], or can act indirectly via stimulating the activity of a major repressor of autophagy, the mammalian target of rapamycin (mTOR) complex 1, through phosphorylation of mTOR and its binding partner, the regulatory associated protein of mTOR (RAPTOR) [48,49]. Consistent with the recent studies in mice [27,44], we also found that the thermogenic stimulus-mediated repression of autophagic activity occurs through a cAMP-dependent PKA pathway in human beige adipocytes (Figure 4B,C). Further analysis is required to reveal the targets of PKA in these cells, and whether it phosphorylates LC3, mTOR, Raptor, ATG13 or other autophagy regulators. PKA also phosphorylates dynamin-related protein 1 (Drp1) at different Ser residues [13,50], which determines its recruitment to mitochondria and subsequently mitochondrial fission in cells [50,51]. In NE-stimulated brown adipocytes, the phosphorylation of Drp1 at Ser600 was shown to promote Drp1-mediated fission [13].

Parkin is one of the major mediators of mitophagy; during the adipogenesis of immortalized 3T3-L1 adipocytes [52] and in differentiating mouse beige [28] and brown [45] adipocytes, its expression was increased. However, the exact role of Parkin in beige adipocytes has not been clarified yet, and contradictory results have been reported. Recently, Cairó et al. found the downregulation of Parkin in response to thermogenic induction in mouse BAT, and the exhibition of high thermogenic activity in Parkin knockout (KO) mice [45]. Moreover, an inefficient beige-to-white adipocyte transition of inguinal WAT was found in PARK2 KO mice [28]. However, Corsa and colleagues published opposing results, claiming that Parkin is not an essential regulator of adipogenesis, differentiation, mitochondrial quality control or the maintenance of beige adipocytes in mice [53]. In our study, we observed that the Parkin-independent (Appendix A) and -dependent (Figure 5A or Figure 5C,D) mitophagy-associated genes were expressed in masked and mature beige adipocytes, and that the genes of the Parkin-mediated pathway were rapidly repressed in response to a thermogenic stimulus. However, the silencing of Parkin did not lead to increased UCP1 or mitochondrial content (Figure 6D,E), so it is likely that Parkin does not play a pivotal role in the observed high mitophagy rate in masked beige cells. On the other hand, we cannot exclude the possibility that a more effective Parkin silencing may result in active beige adipocytes.

The p62 protein could participate in the control of the thermogenic program, since its expression level was increased in response to thermogenic stimulus (Figure 2C,D). In mouse BAT, an adipocyte-specific p62 deficiency resulted in an impaired basal mitochondrial function and response to β-AR stimulation [54]. Moreover, a recent study demonstrated that NDP52 and optineurin play a role in the activation of mitophagy, which is largely independent of Parkin [23], and our data showed that the expression of these genes was significantly reduced as a result of cAMP stimulus in beige adipocytes (Figure 5C,D). In addition to adapter proteins, adipose tissue-specific thermogenic effects have also been described in association with Parkin-independent mitophagy pathways. It was reported that BCL2L13 contributes to beige adipocyte biogenesis [55] and BNIP3 regulates mitochondrial fragmentation in mouse adipocytes [56]. In our experiments, the mRNA level of BCL2L13, FKBP8 and FUNDC1 was decreased as a result of cAMP treatment in white and beige adipocytes (Appendix A), however the BNIP3 and BNIP3L gene expressions changed in the opposite direction (Appendix A). Further investigation is needed to clarify the role of Parkin-independent mitophagy in beige adipocytes.

Our results raise the possibility to obtain thermogenically active beige adipocytes from masked beige cells in vivo only by inhibiting the autophagy/mitophagy process specifically in adipocytes. In mice, the browning process was successfully induced, and BAT thermogenesis was enhanced by the targeted deletion of ATG7, one of the critical regulatory elements in autophagosome formation [57,58,59]. Long-term suppression of autophagy/mitophagy, on the other hand, may result in mitochondrial damage in adipose tissue, and can lead to adverse side effects. Human studies are necessary to confirm the results of mouse experiments, in order to implement procedures for more effective thermogenesis in adipocytes to combat obesity. A high level of autophagy was shown in adipocytes from obese patients [60,61,62]. The upregulation of autophagy-related genes and the accumulation of autophagosomes were demonstrated in the subcutaneous and omental fat depots of obese humans [61,63,64], suggesting that dysregulated autophagy contributes to the pathophysiology of obesity. Recently, it has been shown that in obese adults, lower amounts of activated BAT exist than in lean individuals, but they possess more “brownable” fat [7]. These depots might remain inactive because of the highly active autophagy and mitophagy. Further studies are required to better understand the mechanisms by which physiological mitophagy regulation and its pathologic dysregulation contribute to the cellular homeostasis of beige adipocytes and the development of obesity, respectively.

## 4. Materials and Methods

### 4.1. Materials

The chemicals were obtained from Sigma-Aldrich (Munich, Germany) unless stated otherwise.

### 4.2. Ethics Statement

hASCs were isolated from the subcutaneous abdominal adipose tissue of healthy volunteers, who underwent a planned surgical treatment. Written informed consent from all the participants was obtained before the surgical procedure. The study protocol was approved by the Medical Research Council of Hungary (20571-2/2017/EKU). All the experiments were carried out in accordance with the approved ethical guidelines and regulations.

### 4.3. Isolation, Cell Culture, Differentiation and Treatments of hASC and Simpson–Golabi–Behmel Syndrome (SGBS) Cells

hASCs were isolated from the SVF of abdominal subcutaneous fat as described previously [29]. The absence of mycoplasma was checked by PCR analysis (PCR Mycoplasma Test Kit I/C, PromoCell, Heidelberg, Germany). hASCs and SGBS preadipocytes were seeded in either 6- or 12-well plates (Costar) as per the experiment. Cells were cultured in DMEM-F12 (Dulbecco’s Modified Eagle’s Medium/Nutrient F-12 Ham) medium containing 33 µM biotin, 17 µM pantothenic acid, 100 U/mL penicillin/streptomycin and 10% FBS (Thermo Fisher Scientific, Waltham, MA, USA) at 37 °C in 5% CO_2_ till confluency. This was followed by initiation of 14 day-long differentiation programs. White adipogenic differentiation results in white adipocytes with the presence of masked beige adipocytes [29,30]. Brown adipogenic differentiation of subcutaneous SVF leads to the generation of beige adipocytes; herein, this process is referred to as beige protocol. White adipocyte differentiation was started by using serum-free DMEM-F12 medium supplemented with 33 μM biotin, 17 μM pantothenic acid, 100 U/mL penicillin/streptomycin, 10 μg/mL human apo-transferrin, 20 nM human insulin, 200 pM triiodothyronine, 100 nM cortisol, 2 μM rosiglitazone (Cayman Chemicals, Ann Arbor, MI, USA), 25 nM dexamethasone and 500 μM 3-isobutyl-1-methylxanthine. Four days later rosiglitazone, dexamethasone and 3-isobutyl-1-methylxanthine were omitted from the differentiation medium, which was replaced every fourth day till day 14 [29,30,31]. The brown differentiation protocol was carried out for four days, applying serum-free DMEM-F12 medium containing 33 μM biotin, 17 μM pantothenic acid, 100 U/mL penicillin/streptomycin, 10 μg/mL apo-transferrin, 0.85 μM human insulin, 200 pM triiodothyronine, 1 μM dexamethasone and 500 μM 3-isobutyl-1-methylxanthine. After the fourth day, the medium was changed, and 500 nM rosiglitazone was added while dexamethasone and 3-isobutyl-1-methylxantine were removed. The medium was changed every fourth day and the cells were examined after 14 days of differentiation [29,30,32].

Differentiated adipocytes were subjected to dibutyril-cAMP (500 μM) for 6, 10 or 14 h to induce thermogenesis [65]. CQ (25 μM, 1 h) was used to block the autophagic flux [35,66]. H89 (50 μM, 1 h) was administered primarily to inhibit protein kinase A (PKA).

Transfection and gene silencing experiments were performed using DharmaFECT1 transfection reagent (Dharmacon, T-2001-03) at day 14 of differentiation. White and beige primary adipocytes were incubated with a mixture containing DharmaFECT1 and 50 nM of PARK2-targeted siRNA (Dharmacon, ON-TARGETplus SMARTpool Human Parkin siRNA, L-003603-00-0005), or non-targeted negative control siRNA (Dharmacon, ON-TARGETplus Non-targeting Control Pool, D-001810-10-05) for 4 days in DMEM-F12 medium supplemented with the white or beige differentiation cocktail without 100 U/mL penicillin/streptomycin.

### 4.4. RNA Isolation, RT-PCR, qPCR

Cells were collected in Trizol reagent (Thermo Fisher Scientific), and RNA was isolated manually by chloroform extraction and isopropanol precipitation. RNA concentrations and purity were determined by spectrophotometry. cDNA was generated by a TaqMan™ reverse transcription reagents kit (Thermo Fisher Scientific). The gene primers and probes were designed by Applied Biosystems. Gene expression was determined by qPCR (∆Ct method), as per the described protocols [29,30]. All samples were run in triplicates. Human *GAPDH* was used as an endogenous control. Quantitative PCR experiments were repeated at least five times with SVFs from independent healthy donors, or with SGBS samples from independent passages.

### 4.5. Mitochondrial DNA (mtDNA) Isolation and Quantification by qPCR

Total DNA was isolated by manual phenol–chloroform extraction from samples lysed in Trizol. mtDNA was quantified by qPCR in triplicates on diluted DNA as per the described method [29,30]. Relative mtDNA content was calculated from the difference in the threshold cycle (Ct) values for mtDNA and nuclear specific amplification. Data are expressed as mitochondrial genomes per diploid nuclei. Quantitative PCR measurements were repeated six times with SVFs from independent healthy donors or with SGBS samples from independent passages.

### 4.6. Antibodies and Immunoblotting

Cells were washed once with PBS and collected in 1× Laemmli loading buffer followed by boiling at 100 °C for 10 min. Equal amounts of protein were separated by SDS-PAGE followed by transfer onto PVDF Immobilon-P Transfer Membrane (Merck-Millipore, Darmstadt, Germany). The membranes were blocked by 5% skimmed milk for 1 h, followed by overnight incubation at 4 °C with one of the following primary antibodies: anti-β-actin (1:5000, A2066), anti-LC3 (1:2000, Novus Biological, Centennial, CO, USA, NB100-2220), anti-p62 (1:5000, Novus Biological, NBP1-49956), anti-UCP1 (1:5000, R&D Systems, Minneapolis, MN, USA, MAB6158), anti-total OXPHOS human (1:1000, Abcam, Cambridge, MA, USA, ab110411) or anti-Parkin (1:750, Santa Cruz Biotechnology, Dallas, TX, USA, sc-32282) in tris buffered saline (TBS)-Tween 20 containing 1% skimmed milk solution. HRP-conjugated goat anti-rabbit IgG antibody (1:10,000, Advansta, San Jose, CA, USA, R-05072-500) and HRP-conjugated goat anti-mouse IgG antibody (1:5000, Advansta, R-05071-500) were used as the respective secondary antibodies. Immunoreactive proteins were visualized by Immobilion western chemiluminescence substrate (Merck-Millipore) [30,66]. FIJI software was used for the densitometry.

### 4.7. Immunostaining Analysis

hASCs and SGBS preadipocytes were plated and differentiated in 8-well Ibidi μ-slides. White and beige adipocytes were treated with CQ (25 μM, 1 h), cAMP (500 µM, 8 h) or FCCP (10 μM, 6 h), which served as a positive control for mitophagy. Next, cells were washed with phosphate buffered saline (PBS) and fixed by 4% paraformaldehyde (PFA) for 5 min, followed by permeabilization with 0.1% saponin in 5% skimmed milk dissolved in PBS. Incubations with the primary antibodies anti-TOM20 (1:75, Sigma-Aldrich, WH0009804M1) and anti-LC3 (1:200, Novus Biological, NB100-2220) were kept overnight at room temperature. Secondary antibody incubation was kept for 3 h with Alexa 647 goat anti-mouse IgG (1:1000, Thermo Fischer Scientific, A21236) and Alexa 488 goat anti-rabbit IgG (1:1000, Thermo Fischer Scientific, A11034). Antibodies were applied, and additional washing steps between and after antibody usage were performed in the presence of 0.1% saponin in PBS for effective cell permeabilization. Propidium iodide (1.5 µg/mL) was used to stain the nucleus for 20 min.

### 4.8. Image Acquisition

Images were obtained by confocal laser scanning microscopes from the double labelled (TOM20 and LC3) cells to determine LC3 punctae, mitochondrial fragmentation and co-localization in primary and SGBS adipocytes. An Olympus FluoView 1000 confocal microscope was used in the case of primary adipocytes. For the excitation of Alexa Fluor 488 goat anti-rabbit IgG, the 488 nm line of an Argonion laser was used, while for Alexa Fluor 647 goat anti-mouse IgG a 633-nm He-Ne laser was used, and for propidium iodide a 543-nm He-Ne laser was used. The fluorescence emissions of Alexa Fluor 488 and Alexa Fluor 647 (Thermo Fisher Scientific) were detected through 500–530 nm and 655–755 nm bandpass filters, respectively, while the detection of the fluorescence of propidium iodide was achieved with a 555–625 nm band pass filter. Images were taken in sequential mode to minimize cross-talk between the channels. Images of approximately 1 μm thick optical sections, each containing 512 × 512 pixels (pixel size was ~200 nm), were obtained with a 60× UPLSAPO oil immersion objective (NA 1.35) [67]. In the case of SGBS cells, the images were acquired on a Nikon A1 Eclipse Ti2 confocal laser-scanning microscope by using a Plan Apo 60× water (numerical aperture (NA) 1.27) objective, with a pixel size of 210 nm. Alexa 488 and Alexa 647 were excited at 488 nm and 647 nm, respectively, while propidium iodide was excited at 561 nm. Fluorescence emissions of Alexa Fluor 488 and Alexa Fluor 647 were collected by using 525/50 nm and 700/75 nm bandpass filters, respectively, while detection of the fluorescence of propidium iodide was achieved with a 595/50 nm bandpass filter. Quantification for LC3 punctae and fragmented mitochondria was performed using FIJI software (*n* = 50–60 cells from 3 primary donors; *n* = 10–15 SGBS cells). Co-localization of TOM20 and LC3 was assessed by calculating the Pearson’s correlation coefficients (PCC) between the pixel intensities of the two detection channels [39,40].

### 4.9. Statistical Analysis

Results are expressed as the mean ± SD for the number of assays indicated. To compare two groups, a two-tailed paired Student’s *t*-test was used. The Mann–Whitney U test was also used to determine statistically significant differences. Statistical analysis was performed using the Graphpad Prism 8 software.

## Figures and Tables

**Figure 1 ijms-21-06640-f001:**
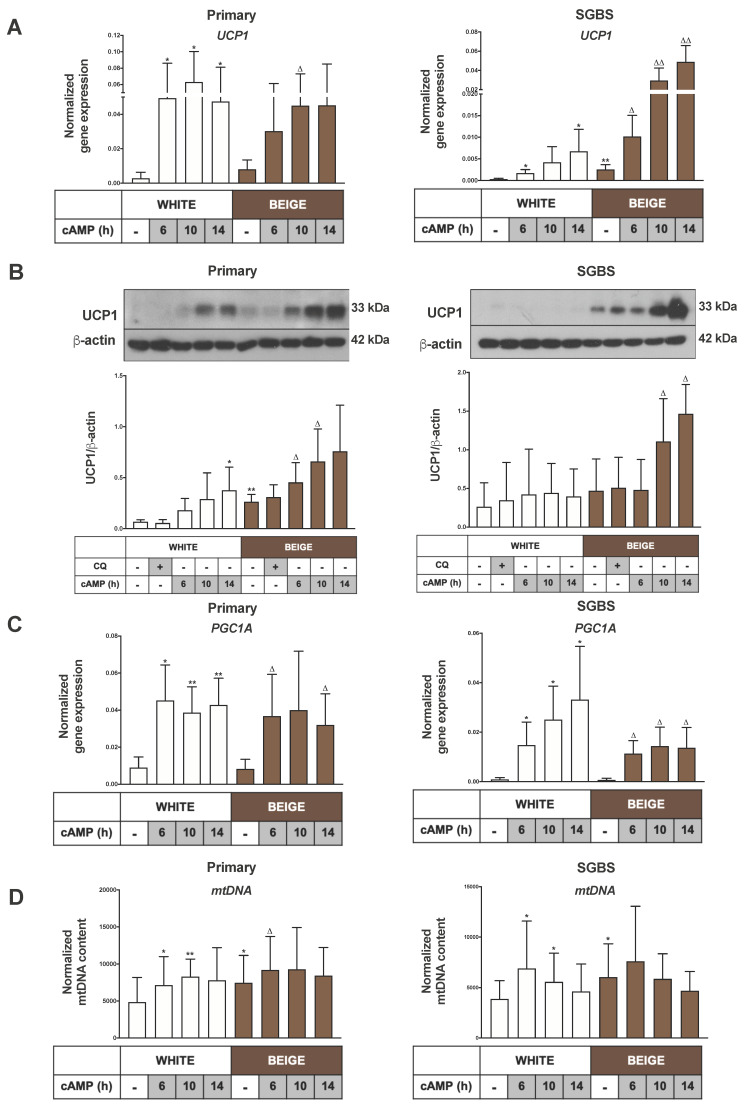
The expressions of Uncoupling protein 1(UCP1), Peroxisome Proliferator-Activated Receptor Gamma Coactivator 1-Alpha (PGC1A) and mitochondrial DNA (mtDNA) were elevated upon cAMP stimulation. Abdominal subcutaneous hASCs and SGBS cells were differentiated to white or beige adipocytes for 2 weeks. A quantity of 500 µM dibutyryl-cAMP was administered for 6 to 14 h to induce thermogenesis, and chloroquine (CQ) treatment (25 µM, 1 h) was applied to block the lysosomal degradation activity. (**A**) UCP1 gene expression in primary (left) and Simpson–Golabi–Behmel syndrome (SGBS) cells (right) normalized to GAPDH, quantified by RT-qPCR (*n* = 5). (**B**) Representative immunoblots and densitometry analysis of UCP1 protein expression in hASC-derived (left) and SGBS cells (right) normalized to β-actin (*n* = 5). (**C**) PGC1A gene expression in primary (left) and SGBS cells (right) normalized to GAPDH quantified by RT-qPCR (*n* = 5). (**D**) Normalized mitochondrial DNA content quantified by qPCR in hASC-derived (left) and SGBS cells (right) (*n* = 6). Results are expressed as mean ± SD. Statistics: two-tailed paired student *t*-test; * *p* < 0.05; ** *p* < 0.01; * represents significant as compared to white untreated sample; △ *p* < 0.05; △△ *p* < 0.01; △ represents significant as compared to beige untreated sample.

**Figure 2 ijms-21-06640-f002:**
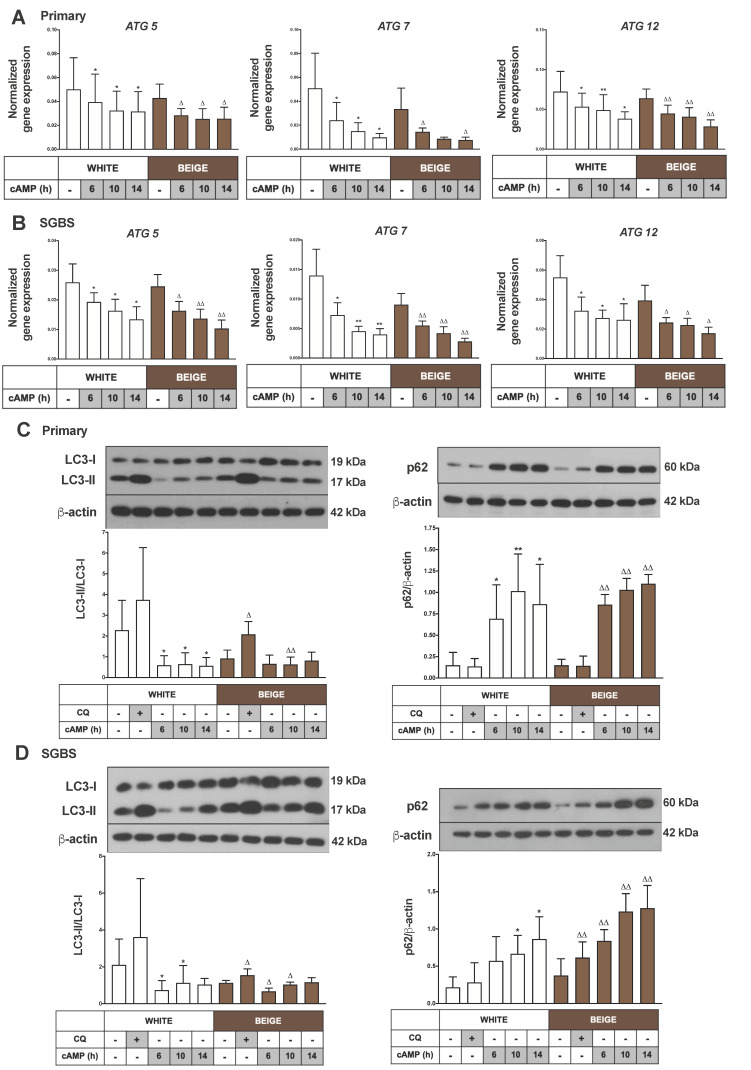
cAMP treatment represses autophagy rate and flux. Adipocytes were differentiated and treated as in Figure 1. Autophagy related (ATG) 5, ATG7 and ATG12 gene expression (**A**) in primary and (**B**) SGBS cells normalized to GAPDH, quantified by RT-qPCR (*n* = 5). Representative immunoblots and densitometry analysis of LC3-II/LC3-I protein expression ratio (left) and p62 protein expression (right) normalized to β-actin (**C**) in hASC-derived and (**D**) in SGBS cells (*n* = 5). Results are expressed as mean ± SD. Statistics: two-tailed paired student *t*-test; * *p* < 0.05; ** *p* < 0.01; * significant as compared to white untreated sample; △ *p* < 0.05; △△ *p* < 0.01; △ significant as compared to beige untreated sample.

**Figure 3 ijms-21-06640-f003:**
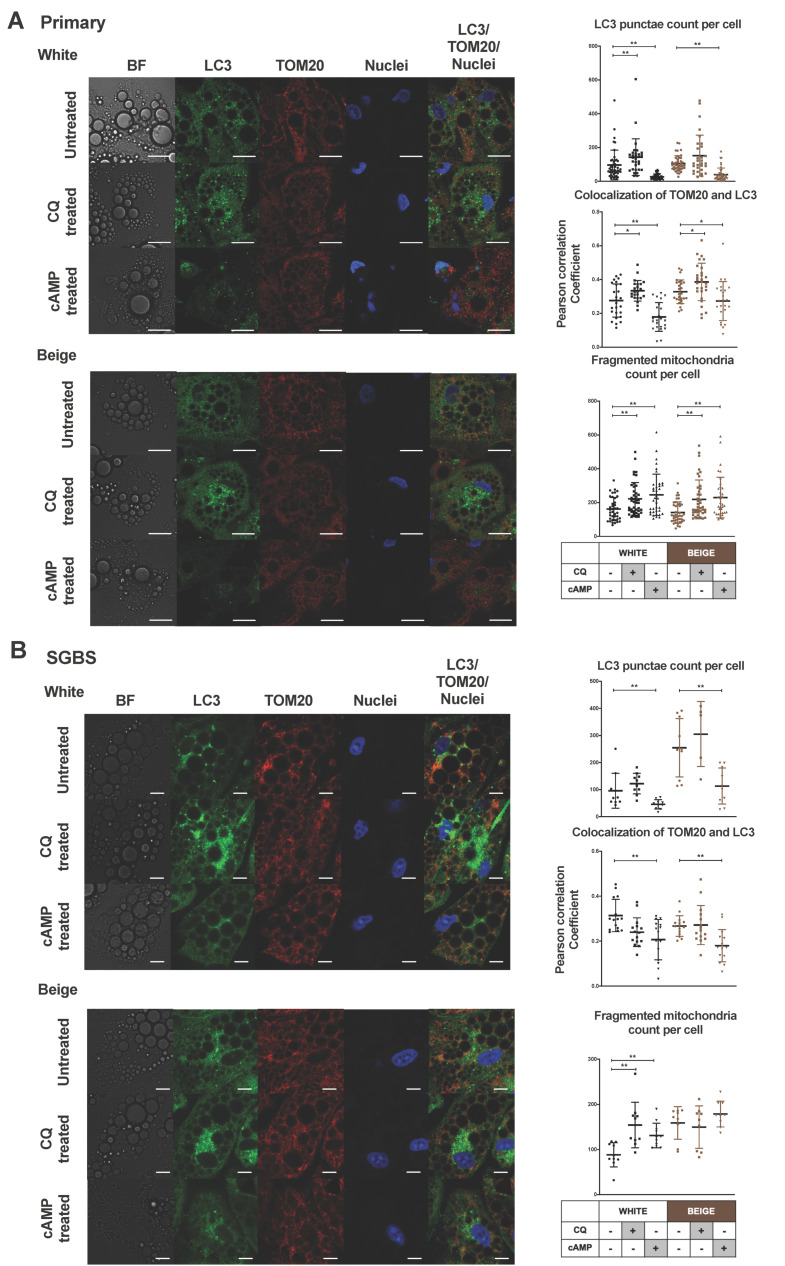
Confocal microscopy images show decreased LC3 punctae, co-localization of LC3 Translocase of outer mitochondrial membrane (TOM) 20 and increased mitochondrial fragmentation in cAMP-stimulated adipocytes. Adipocytes were differentiated as in Figure 1 and Figure 2. Representative confocal microscopy images of LC3 and TOM20 immunostaining (left) under untreated conditions and upon CQ (25 µM, 1 h) and cAMP (500 µM, 8 h) treatment in white and beige (**A**) primary and (**B**) SGBS cells. Scale bars represent 5 µm (**A**) or 10 µm (**B**). Quantification of LC3 punctae, co-localization of LC3-TOM20 and fragmented mitochondria (right) from confocal images (**A**) in hASC-derived (*n* = 40–60 cells, from 3 donors) and (**B**) SGBS cells (*n* = 10–15 cells). Nuclei were stained by propidium-iodide. Results are expressed as mean ± SD. Statistics: Mann–Whitney U test; * *p* < 0.05; ** *p* < 0.01.

**Figure 4 ijms-21-06640-f004:**
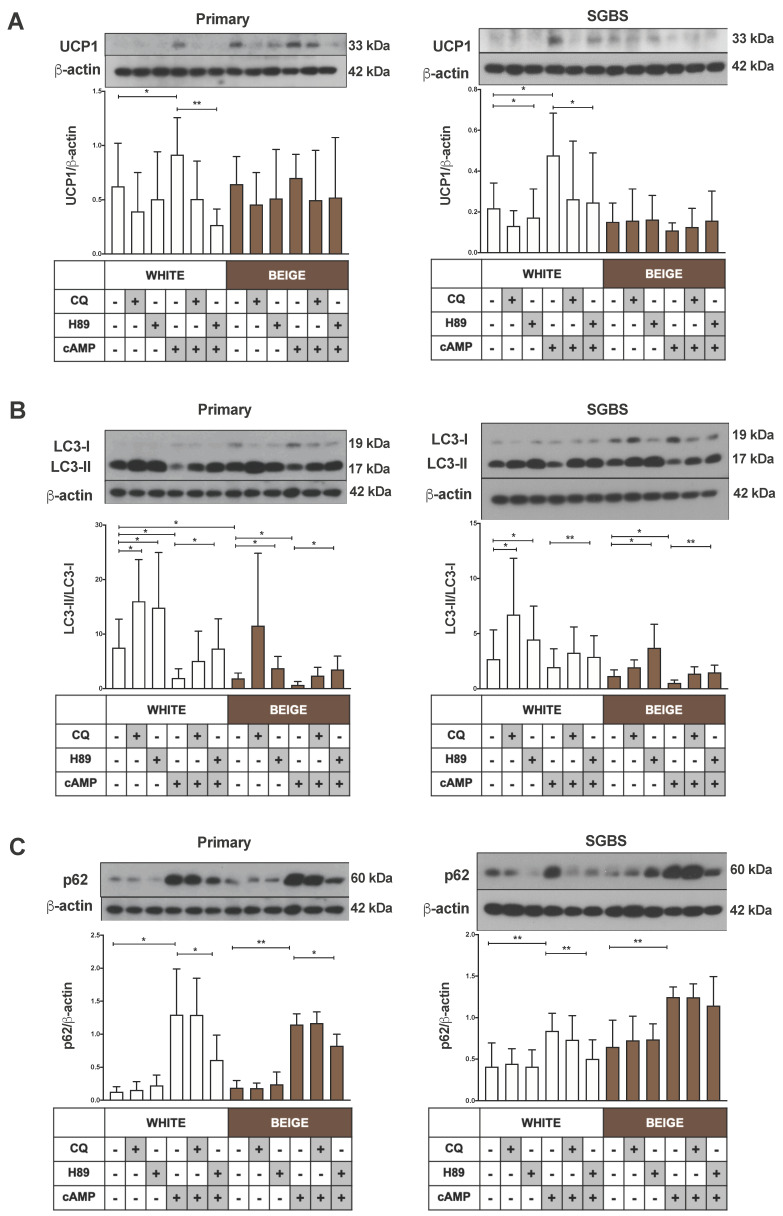
The inhibition of the Protein kinase A (PKA) pathway by H89 alleviates the effect of cAMP on autophagy/mitophagy and thermogenesis. Adipocytes were differentiated as in Figure 1, Figure 2 and Figure 3, and treated with CQ (25 µM, 1 h), H89 (50 μM, 1 h) or cAMP (500 µM, 6 h). Representative immunoblots and densitometry analysis of (**A**) UCP1 (*n* = 4), (**B**) LC3-II/LC3-I ratio (*n* = 5), and (**C**) p62 (*n* = 5) protein expression in primary (left) and SGBS cells (right) normalized to β-actin. Results are expressed as mean ± SD. Statistics: two-tailed paired student *t*-test; * *p* < 0.05; ** *p* < 0.01.

**Figure 5 ijms-21-06640-f005:**
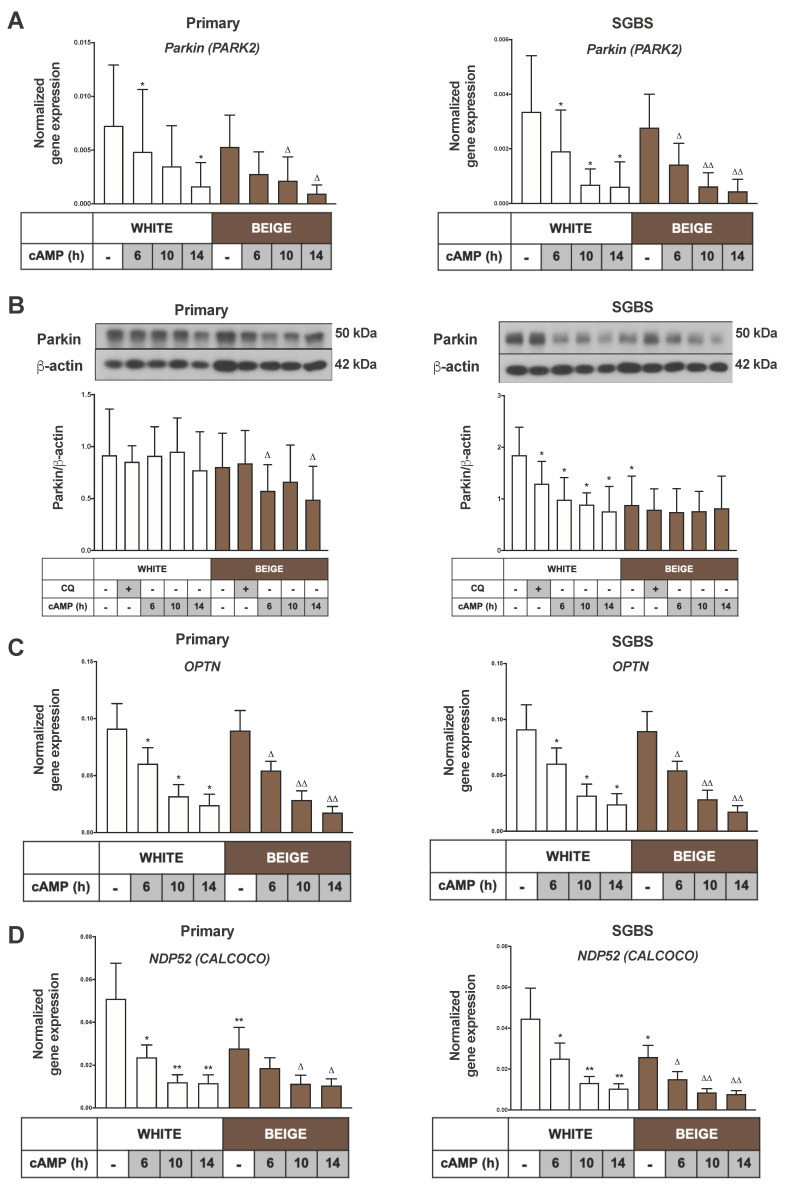
Parkin, Optineurin (OPTN) and Nuclear dot protein (NDP) 52 (CALCOCO) expression is reduced as a result of cAMP treatment. Adipocytes were differentiated and treated as in Figure 1, Figure 2 and Figure 3. (**A**) PARK2 gene expression in primary (left) and SGBS cells (right) normalized to GAPDH, quantified by RT-qPCR (*n* = 5). (**B**) Representative immunoblots and densitometry analysis of Parkin protein expression in hASC-derived (left) and SGBS cells (right) normalized to β-actin (*n* = 5), (**C**) OPTN and (**D**) NDP52 (CALCOCO) gene expression in primary (left) and SGBS cells (right) normalized to GAPDH, quantified by RT-qPCR (*n* = 5). Results are expressed as mean ± SD. Statistics: two-tailed paired student *t*-test; * *p* < 0.05; ** *p* < 0.01. *: significant as compared to white untreated sample. △ *p* < 0.05; △△ *p* < 0.01. △: significant as compared to beige untreated sample.

**Figure 6 ijms-21-06640-f006:**
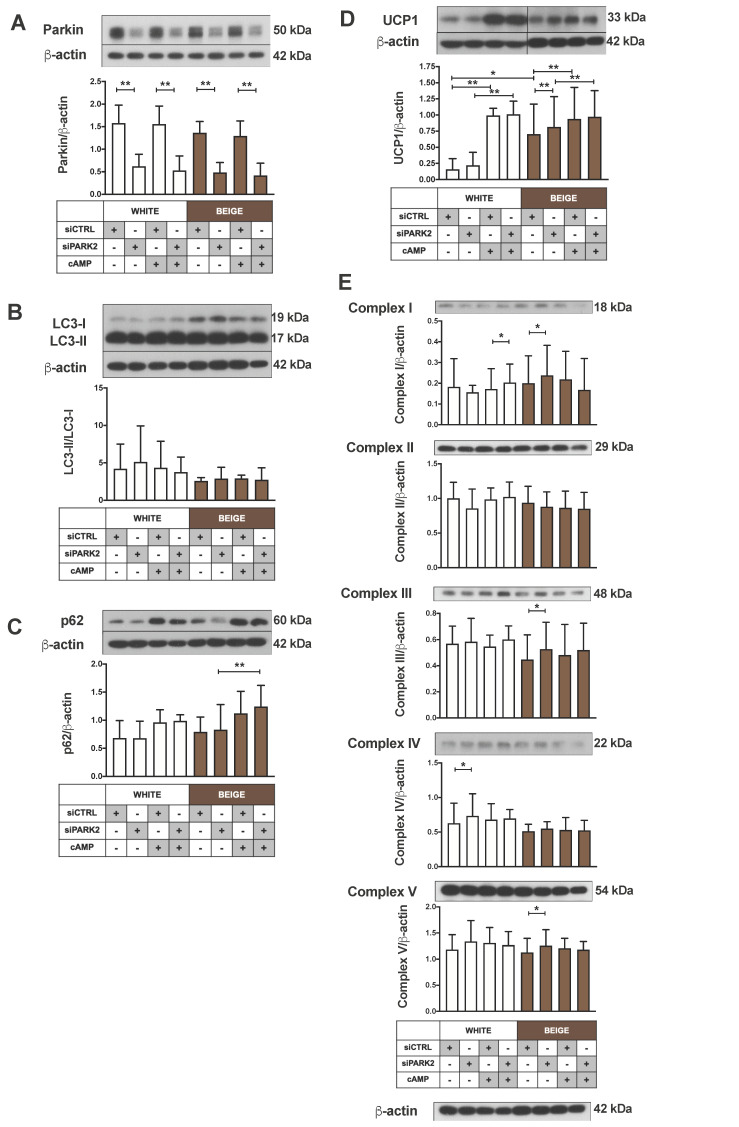
The impact of Parkin silencing on autophagy and the expression of UCP1 and mitochondrial complex subunits. Representative immunoblots and densitometry analysis of (**A**) Parkin, (**B**) LC3-II/LC3-I ratio, (**C**) p62, (**D**) UCP1 and (**E**) mitochondrial complex subunits protein expression in PARK2-silenced hASCs under untreated conditions or after cAMP-stimuli (500 µM, 10 h), normalized to β-actin (*n* = 4). Results are expressed as mean ± SD. Statistics: two-tailed paired student *t*-test; * *p* < 0.05; ** *p* < 0.01.

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
