# Peer review of "Thermogenic Activation Downregulates High Mitophagy Rate in Human Masked and Mature Beige Adipocytes"

_ijms, 2020, doi:10.3390/ijms21186640_

Round 1

Reviewer 1 Report

Szatmári-Tóth et al. show that in human white and beige adipocytes, cAMP, in addition to stimulating classic indictors of thermogenesis, also inhibits standard markers of autophagy, including ATG5/7/12 genes, and LC3-II formation. They further show, using confocal imaging, that mitophagy is decreased following thermogenic stimulation by camp. PKA appears to be involved in the decreased mitophagy, as PKA inhibition alleviates some of the indictors of  cAMP mediated mitophagy repression. To determine if inhibition of auto/mitophagy is sufficient to induce thermogenesis, the authors knockdown PARK2, finding that this perturbation has only minimal effects on various thermogenic indictors.     The manuscript is purely descriptive, showing that markers of autopagy are inhibited with cAMP  stimulation. The results of the sole mechanistic examination, that of PKA inhibition, are uncompelling, leading to little insight. Thus, the manuscript overall is not found to be possess a high level of novelty, although the fact that it is being done in human adipocytes does add to that component. Otherwise, the results and interpretation are deemed to be acceptable. The manuscript, as stands, is also replete with quite a few fundamental grammatical errors and needs to be carefully edited. 

Author Response

We thank the Reviewer for the positive comments and useful suggestions which helped to improve the quality of the manuscript. We have also performed a thorough check for any grammatical errors and improved the English in certain parts of the manuscript.

Reviewer 2 Report

This manuscript aimed to study the effects of thermogenic activation in human masked and mature beige adipocytes. The authors found increased expression of UCP1, mitochondrial DNA content and respiratory chain complex subunits under cAMP-induced thermogenic activation. In addition, they also characterized reduced parkin-dependent mitophagy. The conclusion this manuscript reached is that mitophagy is repressed upon short-term adrenergic stimulus, as a fast regulatory mechanism to provide high mitochondrial content for thermogenesis. The manuscript is organized well and should be interesting to the readers in related research fields.

Major concern

Previous paper indicated that mitochondrial fission is prominently inhibited Cyclic AMP-dependent Protein Kinase Phosphorylation of Drp1 at Ser637. This would result in mitochondria hyperfusion. The authors’ results are contradicted the PKA dependent regulation of mitochondria dynamics.

Minor concerns

  1. To manipulate cAMP level and PKA in cells, Forskolin and IBMX are potential chemicals to apply.
  2. The authors should provide more information about how they decided to examine 6, 10 and 14 hr samples, but not 2 and 4 hr in Figure 1.
  3. The mitochondria network morphology is difficult to see clearly in the figures. Better images are highly recommended.

Author Response

We thank the Reviewer for the positive comments and useful suggestions which helped to improve the quality of the manuscript.

Answer to the major concern: PKA and other kinases can phosphorylate Drp1 at different Ser residues that determine its recruitment to mitochondria and subsequently mitochondrial fission in cells. We are aware that phosphorylation at Ser637 inhibits Drp1 GTPase activity which promotes mitochondrial fusion (ref. 53 was added). This was shown originally in transfected HeLa cells. Recently, Yu et al. used HEK 293T cells and drew opposite conclusions (PMID: 31533986). However, in NE-stimulated brown adipocytes, phosphorylation of Drp1 at Ser600 by PKA was shown to promote Drp1-mediated fission (ref. 13). This phosphorylation event also resulted in mitochondrial fragmentation in HeLa cells (ref. 54 was added). The target Ser residue of PKA on Drp1 might differ in distinct cell types and might be determined by currently unrevealed signaling events that need to be further investigated. Based on the comment of the Reviewer, the Discussion was extended between lanes 377-380.

Question 1: To manipulate cAMP level and PKA in cells, Forskolin and IBMX are potential chemicals to apply.

Answer 1: In our experiments, we have used a dibutylated form of cAMP (based on ref. 33) which is membrane permeable and hence can have a direct effect on the downstream effectors of the PKA pathway. Applying other chemicals is considered in future studies.

Question 2: The authors should provide more information about how they decided to examine 6, 10 and 14 hr samples, but not 2 and 4 hr in Figure 1.

Answer 2: Generally, 2 to 4 hrs long cAMP treatments are used to investigate changes in gene expression or oxygen consumption (e.g. ref. 33). Since we mainly focused on changes at protein or at morphological levels, we decided to apply longer treatment periods and investigate the time dependent effects of the compounds.

Question 3: The mitochondria network morphology is difficult to see clearly in the figures. Better images are highly recommended.

Answer 3: We included representative high-resolution confocal microscopy images of TOM20 immunostaining as Supplementary figure 3 in the revised Supplementary file.

Reviewer 3 Report

In this study, Szatmári-Tóth and colleagues addressed key physiological processes surrounding adipocyte mitochondrial biology. They found out that cAMP stimulation resulted in exacerbated mitochondrial activity as assessed by measuring expression levels of key proteins including UCP1 and some (but not all) respiratory chain components as well as decreased mitophagy. They also found out that upon cAMP stimulation, a fragmented phenotype was evident in mitochondria from both white and beige adipocytes in a PKA-dependent axis. This study provides very interesting data and should become further considered for publication. I do have a few concerns. 1) Please consider performing two-way Anova with Scheffe’s post-hoc test instead of t tests when assessing the effects several different variables including cell type in the absence or presence of CQ, cAMP and/or H89, etc. If you find some changes, please conclude accordingly. 2) Please consider including in Fig. 3 image close-ups for the TOM20 channel so that enhanced mito fragmentation is evident for the reader.

Author Response

We thank the Reviewer for the positive comments and useful suggestions which helped to improve the quality of the manuscript.

Question 1: Please consider performing two-way Anova with Scheffe’s post-hoc test instead of t tests when assessing the effects several different variables including cell type in the absence or presence of CQ, cAMP and/or H89, etc. If you find some changes, please conclude accordingly.

Answer 1: In our study, we drew conclusions based on the comparison of two groups. We compared treated white adipocytes to untreated ones, treated beige adipocytes to untreated ones, etc. In this case, we think that the pair wised t-tests are an acceptable way of the statistical analysis.

Question 2: Please consider including in Fig. 3 image close-ups for the TOM20 channel so that enhanced mito fragmentation is evident for the reader. 

Answer 2: We included representative high-resolution confocal microscopy images of TOM20 immunostaining as Supplementary figure 3 in the revised Supplementary file.